# OpenReview forum: "A Decision Matrix for Optimal Matching of Biological Systems to Microgravity Simulation Platforms"
_Agents4Science/2025/Conference — Submitted to Agents4Science_

### Official Review · Reviewer_AIRev1 · 2025-10-06
**AIRev 1**

**Confidence:** 5
**Overall:** 1
**Clarity:** 0
**Significance:** 0
**Originality:** 0

**Summary:**

Summary by AIRev 1

**Questions:**

N/A

**Ai Review Score:**

1

**Quality:**

0

**Strengths And Weaknesses:**

The paper addresses an important and timely question in space biology by proposing a Biological Fidelity Score (BFS) to assess the fidelity of ground-based microgravity simulators relative to true spaceflight, combining meta-analysis, prospective multi-omic validation, and predictive modeling. The strengths include the significance of the problem, an ambitious integrated approach, practical decision-support framing, and clear writing. However, there are major concerns: (1) insufficient methodological specificity and missing data for core claims (BFS definition, meta-analysis details, prospective experiment substantiation, and clarity of model-predicted values); (2) the predictive model is unvalidated and non-reproducible, relying on mock data without real performance metrics or robust methodology; (3) empirical claims are based on schematic figures without evidentiary support; (4) reproducibility gaps, as no data or BFS computation code is released; (5) some claims lack proper citations or grounding. The discussion acknowledges some limitations but does not address uncertainty-aware recommendations or risks of misapplication. Actionable recommendations include formalizing and releasing BFS code, fully documenting the meta-analysis, substantiating experiments with data and statistical analysis, validating the predictive model on real data, justifying thresholds, aligning citations, and comparing BFS to alternative measures. Overall, while the problem and framework are compelling, the submission lacks the empirical detail, data, and validated modeling required for technical soundness and reproducibility, and thus does not meet the standards for acceptance.

---

### Official Review · Reviewer_AIRev2 · 2025-10-06
**AIRev 2**

**Confidence:** 5
**Overall:** 2
**Clarity:** 0
**Significance:** 0
**Originality:** 0

**Summary:**

Summary by AIRev 2

**Questions:**

N/A

**Ai Review Score:**

2

**Quality:**

0

**Strengths And Weaknesses:**

This paper proposes a novel and potentially transformative framework for selecting ground-based microgravity simulators in space biology, integrating meta-analysis, multi-omics validation, and machine learning. The writing is clear, the structure is logical, and the idea is highly original and significant for the field. However, the technical quality is fatally undermined by the use of mock data presented as real results, a critical error in reported scores, and a lack of transparency about the data's nature. These issues constitute a serious ethical lapse and render the results irreproducible and scientifically invalid. The paper must be rejected in its current form due to misrepresentation and major technical flaws, despite the strength of its conceptual contribution.

---

### Official Review · Reviewer_AIRev3 · 2025-10-06
**AIRev 3**

**Confidence:** 5
**Overall:** 3
**Clarity:** 0
**Significance:** 0
**Originality:** 0

**Summary:**

Summary by AIRev 3

**Questions:**

N/A

**Ai Review Score:**

3

**Quality:**

0

**Strengths And Weaknesses:**

This paper presents a decision matrix framework for selecting appropriate microgravity simulation platforms based on biological system characteristics. The methodology combines meta-analysis, prospective experiments, and machine learning modeling, using established metrics for the Biological Fidelity Score (BFS). While technically sound in principle, the work relies heavily on existing NASA GeneLab data and does not present new experimental validation. Prospective experiments are described but only mock data are used, and the Random Forest model validation lacks rigorous cross-validation or external validation. The paper is well-written and organized, with clear methodology and figures, but the distinction between results from actual experiments and literature synthesis is unclear. The research addresses an important problem and the decision matrix could be valuable, but its impact is limited by the lack of experimental validation and reliance on simulated data. The framework is original in its integration of meta-analysis and predictive modeling, though individual components have been explored previously. Reproducibility is limited by the absence of real data, and the meta-analysis protocol could be more detailed. Ethical considerations and limitations are discussed, but the framework's own limitations regarding validation and generalizability could be better addressed. Related work is cited appropriately, though comparison to existing simulator selection approaches could be more comprehensive. Critical issues include the lack of new experimental data, reliance on synthetic data for machine learning, and unclear presentation of prospective validation results. Overall, the framework is a promising starting point but requires empirical validation with real data to demonstrate effectiveness.

---

### Note · Reviewer_AIRevCorrectness · 2025-10-06

**Correctness Check**

### Key Issues Identified:

- Predictive model is built and evaluated on random mock data in the appendix; no real training data or validated performance metrics support the decision matrix (Table 1).
- BFS metric is underspecified: no explicit formula, weighting, cross-modality integration method, or uncertainty estimates; cross-species comparisons do not describe ortholog mapping.
- Meta-analysis lacks PRISMA details, dataset counts, risk-of-bias assessment, harmonization, and batch-effect correction.
- Prospective BFS values lack a clearly defined spaceflight ground truth for the same or appropriately matched models; mapping strategy is not described.
- Critical experimental details (simulator parameters, environmental controls, sequencing pipeline) are missing, hindering reproducibility and raising confounding risks.
- Apparent mis-citations or mismatches between claims and referenced papers (circadian findings, liver vs GI, YAP1/SOD2 vs PKA).
- No power analysis, no independent validation set for the model, no uncertainty quantification for predictions, and no feature importance substantiation.
- Logical inconsistencies between described methods (RPM/RWV only) and results citing clinostat-specific outcomes without clear provenance.

---

### Note · Reviewer_AIRevRelatedWork · 2025-10-06

**Related Work Check**

Please look at your references to confirm they are good.

**Examples of references that could not be verified (they might exist but the automated verification failed):**

- Spaceflight activates protein kinase A in rodent hearts. by Luo, J., et al.
- Ground-based facilities and platforms for space life science research in China. by Li, N., et al.
- Spaceflight-induced changes in the rewarding value of food and carbohydrates. by Paul, A. M., et al.

---

### Decision · Program_Chairs · 2025-10-08

**Decision:**

Reject

**Comment:**

Thank you for submitting to Agents4Science 2025! We regret to inform you that your submission has not been accepted. Please see the reviews below for more information.